# Effects of Alfalfa Crop Rotation on Soil Nutrients and Loss of Soil and Nutrients in Semi-Arid Regions

**Ang Li [1], Yingzhen Wu [2], Xisheng Tai [1], Suzhen Cao [1] and Tianpeng Gao [1,3,*]**

[1] Engineering Research Center of Mining Pollution Treatment and Ecological Restoration of Gansu Province, College of Urban Environment, Lanzhou City University, Lanzhou 730070, China; liang691224@163.com (A.L.)

[2] College of Marxism, Gansu Agricultural University, Lanzhou 730070, China

[3] College of Biological and Environmental Engineering, Xi'an University, Xi'an 710065, China

* Correspondence: zkgtp@163.com

**Abstract:** Soil desertification is an important factor leading to soil degradation and environmental problems such as atmospheric or water pollution. Conservation agriculture, such as crop rotation and conservation tillage, can reduce soil erosion and nutrient loss caused by wind in semi-arid regions. However, the relationships between the loss of soil and nutrients and surface vegetation characteristics are frequently obtained according to a short-term simulation experiment, which makes the application of the conclusions limited. In this study, we conducted a 4-year field experiment continuously with three crop rotations, i.e., spring wheat (*Triticum aestivum* L.) cropping, alfalfa (*Medicago sativa* L.) cropping, and fallow when previous rapeseed crops (*Brassica napus* L.) were being harvested; measured the surface vegetation characteristics, soil nutrients, and loss of soil and nutrients caused by wind; and analyzed their variations and quantitative relations. The findings of this study indicated that the coverage, height, and biomass of the aboveground vegetation in three rotations in the spring and autumn had significant differences, and the rank order was fallow field < wheat field < alfalfa field. With the extension of growing years, the soil organic carbon (SOC) and total nitrogen (TN) of a 0–5 cm soil layer all increased to varying degrees and had significant differences among the rotations during the late stages of the experiment ($p < 0.05$), while the changes in the total phosphorus (TP) and total potassium (TK) were small, and their values had decreasing trends. The available nitrogen (AN) and available phosphorus (AP) decreased first and then increased during the experimental period, while the available potassium (AK) had an increasing tendency. The increase in soil nutrients in the alfalfa field was the highest during the whole experiment period, while the loss of soil and nutrients (SOC, TN, TP, and TK) was the lowest, followed by the wheat and fallow fields. There were significant negative correlations between the surface vegetation characteristics and the loss of soil and nutrients ($p < 0.01$), while the correlations between soil loss and the loss of soil nutrients were significantly positive. In summary, alfalfa crop rotation can obviously reduce the loss of soil and nutrients in semi-arid areas, which is conducive to the sustainability of agroecosystems.

**Keywords:** sustainability of agroecosystems; soil organic carbon; soil wind erosion; crop rotation

## 1. Introduction

Soil is the material base of agroecosystems that supports the growth of crops and determines their quality and yield [1–3]. Traditional agricultural practices can cause water shortages, soil loss, and fertility reduction, which ultimately lead to the degradation of agroecosystems [4–6]. To improve the sustainability of agroecosystems, more attention has been paid to the transformation from traditional cropping to conservation agriculture, such as no-tillage agriculture and crop rotation [7–9]. Crop rotation usually introduces several crop species, increases the soil organic carbon (SOC) content and soil microbial communities, and improves the soil environment compared to the continuous cultivation of one crop species [10,11]. Deiss et al. [12] found that when using no-tillage practices,

the soil organic matter (SOM) of a 0–20 cm soil layer in corn–forage–forage rotation was 12% and 22% higher than that of continuous corn and corn–soybean rotations in silt loam soil; the SOM of a 0–17.5 cm soil layer in corn–forage–forage rotation increased by 11% compared to that of corn–soybean rotation and was similar to that of continuous corn rotation (−0.9%) in clay loam soil. Yang et al. [11] found that after 3-year rotations, the SOC, total nitrogen (TN), and biodiversity of a 0–15 cm soil layer under rice–tiny vetch and rice–fallow rotations increased significantly, and the nutrient structure tended to be more complex when compared to rice–wheat and rice–rape rotations.

Previous studies have confirmed that agricultural intensification and land use changes are the main causes of global biodiversity reduction and habitat degradation [13–15]. Studies on farmland protection have often focused on a single problem, such as soil salinity and alkali [16,17], soil desertification [18–20], and soil fertility reduction [12,21,22]. However, soil degradation in semi-arid irrigated regions is often affected by multiple factors, such as secondary soil salinization, wind erosion, and soil nutrient decline [2,12,19]. To date, few studies have focused on farmland degradation caused by multiple factors. In addition, surface vegetation has a significant impact on soil salinity and nutrients in semi-arid areas, while their quantitative relationships are rarely involved in previous studies in the literature [21,23].

Soil degradation and quality decline are affected by many natural and anthropogenic factors [2,24,25]. At present, the salt-affected land in China is roughly $3.6 \times 10^7$ hm$^2$ and still has an increasing tendency; more than 50% of irrigated cropland is influenced by over-salting, resulting in salinization aggravation and nutrient decline [26]. Wind erosion threatens 50% of China's terrestrial land, which is severe in farmlands in arid and semi-arid irrigated areas, with the mean annual soil loss reaching 3.5 kg·m$^{-2}$, resulting in soil desertification and nutrient reduction [20,25,27,28]. To improve the soil quality, many conservation practices, such as crop rotation, fallow, and no-tillage practices, are being widely popularized in northern China. However, there are relatively few comparative studies on the effects of these measures on soil salinity, wind erosion, and nutrients in semi-arid irrigated areas.

According to our previous study [29], fallowing or spring crops with relatively short growth seasons in semi-arid irrigation regions could result in higher topsoil salinity in the spring. Covering the ground with vegetation and stubble has an obvious effect on the prevention and control of secondary soil salinization. What are the effects of planting spring-sown crops, alfalfa, and fallow on soil nutrients and the loss of soil nutrients? We first hypothesized in this study that the soil nutrient contents and the loss of soil and nutrients can generate different changes in various crop rotations in semi-arid regions, and then we conducted a 4-year field experiment with different crop rotations to (a) explore the responses of vegetation characteristics, soil wind erosion, and nutrient contents to various crop rotations; (b) identify the changes in the vegetative characteristics, nutrient contents, and loss of soil and nutrients under different crop rotations; and (c) reveal the relations between the vegetative characteristics, nutrient contents, and loss of soil and nutrients.

## 2. Materials and Methods

### 2.1. Study Sites

The field experiment was conducted in the Qinwangchuan irrigated region (longitude, 103°30′–103°45′ E; latitude, 36°26′–36°47′ N; and altitude, 1700–2300 m) located in Yongdeng County of Gansu Province, China. The elevation is lower in the south and higher in the north, and the distance is only 25 km between the north and the Tengger Desert. The region is characterized by a semi-arid continental climate, with a mean annual temperature, precipitation, evaporation, and wind speed of 6.2 °C, 287 mm, 1888 mm, and 2.5 m·s$^{-1}$, respectively [29]. The precipitation is uneven monthly, mainly occurring from July to September. The maximum wind speed reaches 20 m·s$^{-1}$ in spring. The soil is classified as calcisols with a bulk density of 1.29 g·m$^{-3}$ and a particle size distribution of 41.7% sand, 48.2% silt, and 9.8% clay in a 0–20 cm soil layer [23]. To avoid the influence of surface

structures on wind erosion, this study was conducted on farmlands near Xiagushan village in the north [29]. The former crop in the experimet field was rapeseed (*Brassica napus* L.).

### 2.2. Experiment Design and Establishment

The field trial began in March 2016. A randomized complete block design with four replicates was applied. The size of each plot was 24 m$^2$ (6 m × 4 m) with a 0.5 m wide border between adjacent plots. The treatments were spring wheat (*Triticum aestivum* L.) cropping, alfalfa (*Medicago sativa* L.) cropping, and fallow (i.e., bare land) when previous rapeseed crops were being harvested. To reduce the influence of external factors on soil nutrients, no fertilizers were applied throughout the experiment period. In March 2016, the selected field was first cleared of weeds, the soil was raked smooth with spike-tooth harrow, and then divided into 12 plots according to the trial design. Wheat and alfalfa were seeded in drills at seeding rates of 30 and 3.1 g·m$^{-2}$, row spacing of 20 cm, and planting depths of 4 and 1 cm, respectively; herbicides were sprayed on the surface of the fallow field [29]. According to the water requirements of wheat during growth, the trial plots were flooded with irrigation in early May, June, and July. Other management practices, such as pest, weed, and disease control, were performed according to local practices. Alfalfa was cut promptly when the plant height exceeded 80 cm, that is, in July and September of the first year, and early June, late July, and early September in other years. In late July and October, the wheat and alfalfa were harvested with stubble of 10 cm retained on the surface, respectively. Wheat was reseeded in the same plot in the spring of the remaining three years, while alfalfa, as a perennial forage grass, was not reseeded.

### 2.3. Sampling Processing and Analyzing

The coverage, height, and biomass of the aboveground vegetation in the plots were determined before wheat was seeded and wheat and alfalfa were harvested, i.e., in early March, late July, and late October [30]. Then, five sample points in an S-shaped pattern were selected in each plot, and the soil under the sample points was obtained by soil auger at four depths of 0–5, 5–10, 10–20, and 20–40 cm and mixed into four samples [29]. Therefore, 144 soil samples were collected (3 times × 4 layers × 12 plots) to analyze their physical and chemical properties each year according to the standard procedures described by Su et al. [31] and Bao [32] using soil extract solutions (soil:water ratio of 1:5). SOC was determined by potassium dichromate oxidation and the oil bath heating method; TN was determined by the semi-micro Kjeldahl method; available nitrogen (AN) was determined by the alkaliolytic diffusion method; total phosphorus (TP) was determined by the molybdenum-antimonic resistance colorimetric method; available phosphorus (AP) was determined by the sodium bicarbonate method; total potassium (TK) and available potassium (AK) were determined by flame spectrophotometry.

To measure soil loss from wheat harvest to crop seedlings of the following year, a rectangular stainless-steel tray (230 cm × 160 cm × 5 cm) with 500 g dried soil from the experimental field was placed in the middle of each plot at the beginning of each month, and the dried weight of the residual soil was measured at the end of each month [33]. The monthly and total soil loss in each plot can be calculated with the following formulas:

$$SL_{i,j} = \frac{\sum_j (MOS_{i,j} - MRS_{i,j})}{A} \tag{1}$$

$$SL_i = \sum SL_{i,j} \tag{2}$$

where $SL_{i,j}$ is the soil loss of the i th plot (where i = 1 to 12) in the j th month (where j = September to June of the following year), MOS is the mass of the original soil in the tray, MRS is the mass of the residual soil, A is the area of the tray, and $SL_i$ is the soil loss of the i th plot from wheat harvest to crop seedlings of the following year.

### 2.4. Statistical Analysis

All statistical analyses were performed with Excel 2010 and SPSS 20. The differences in the treatments were analyzed using one-way analysis of variance (ANOVA), and multiple comparisons were performed using Duncan's multiple range tests at a significance of 0.05 [29,31]. Correlational relationships among the measured factors in the field experiment were analyzed using Pearson's method.

## 3. Results

### 3.1. Changes of Surface Vegetation Properties under Three Crop Rotations

Crop rotations affected the surface vegetation characteristics. As shown in Figure 1a, the vegetation coverage in the wheat field increased from March to July and reached its maximum in July, varying from 36.3% to 82.5%. After the wheat harvest (from early August) to March of the next year, the coverage decreased continuously due to wind and rain and varied from 5.0% to 14.0% in October and 5.0% to 8.3% in March, respectively. The coverage of alfalfa fields also showed an increasing trend from March to July and varied from 80.0% to 97.5% in July; the coverage fluctuated from 66.3% to 91.3% in October and 25.0% to 33.8% in March because of being mowed and harvested. There was a significant difference in vegetation coverage between the wheat and alfalfa fields ($p < 0.05$). Similar to the vegetation coverage, Figure 1b shows that the vegetation height varied from 7.0 to 8.0 cm in March, 35.0 to 84.0 cm in July, 7.0 to 11.0 cm in October in the wheat field, and 8.2 to 13.5 cm in March, 37.5 to 107.0 cm in July, and 14.3 to 37.8 cm in October in the alfalfa field, respectively; the difference in vegetation height between the wheat and alfalfa fields was also significant. Figure 1c shows that the surface vegetation biomass varied from 0.01 to 0.06 kg·m$^{-2}$ in March, 0.27 to 1.29 kg·m$^{-2}$ in July, and 0.01 to 0.21 kg·m$^{-2}$ in October in the wheat field, and 0.05–0.11 kg·m$^{-2}$ in March, 0.21–0.62 kg·m$^{-2}$ in July, 0.1–0.3 kg·m$^{-2}$ in October in the alfalfa field, respectively; there was a significant difference in the biomass between the wheat and alfalfa fields. The plants in the fallow field were negligible due to herbicides and manual weeding. Altogether, the coverage, height, and biomass of the aboveground vegetation in three crop rotations in March and October had significant differences and were ranked as alfalfa field > wheat field > fallow field.

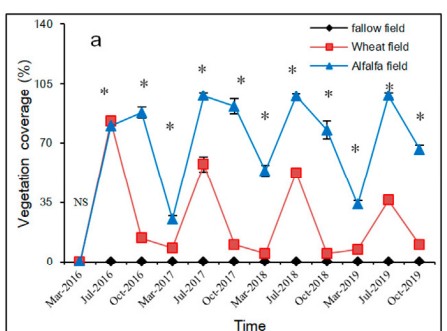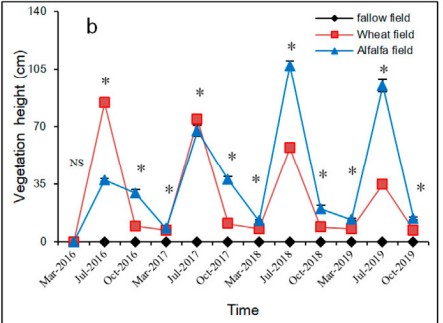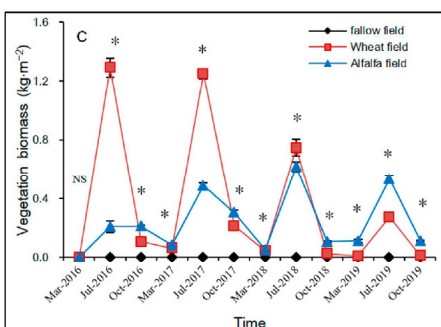

**Figure 1.** Changes in surface vegetation characteristics under the three crop rotations. (**a**) shows the changes in vegetation coverage under the three crop rotations. (**b**) shows the changes in vegetation height under the three crop rotations. (**c**) shows the changes in vegetation biomass under the three crop rotations. Asterisks and NS indicate significant differences ($p < 0.05$) and insignificant differences between treatments, respectively.

### 3.2. Changes in SOC, TN, TP, and TK under Three Crop Rotations

Crop rotations affected surface vegetation and further influenced soil nutrients. Figure 2a shows that the SOC in topsoil (0–5 cm) increased with the extension of growing years, added by 21.9%, 33.1%, and 64.5% in the fallow, wheat, and alfalfa fields at the end of the experiment compared to that at the beginning of the experiment, and the differences among the rotations changed from no significance in the early stage to significance in the later stage. Analyzing the soil profile, the SOC change in different soil layers was small

at the beginning, and the differences among the rotations were not significant (Figure 2b). While at the end (Figure 2c), the SOC in the upper layer was larger than that in the lower layer, and the SOC in a 0–5 cm soil layer in the fallow, wheat, and alfalfa lands increased by 14.4%, 30.7%, and 30.0% compared with that of a 20–40 cm layer; the SOC of topsoil increased by 11.6% and 11.5% in the wheat and alfalfa fields compared to that in the fallow fields, and showed significant difference among the rotations.

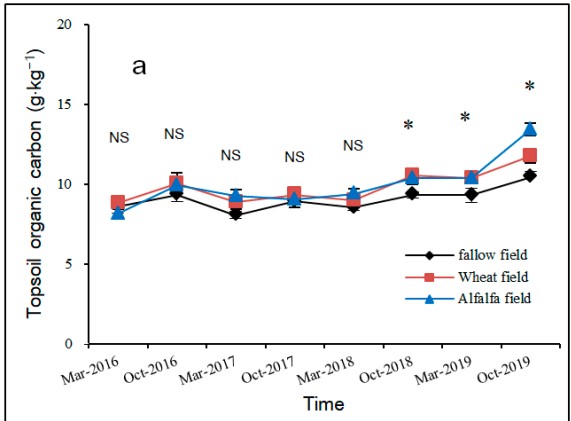
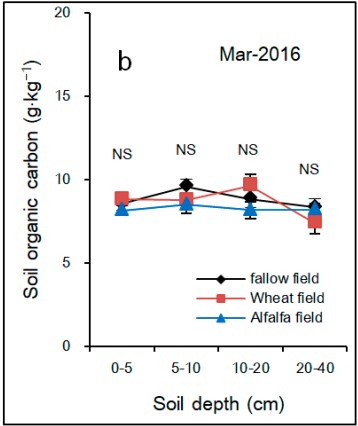
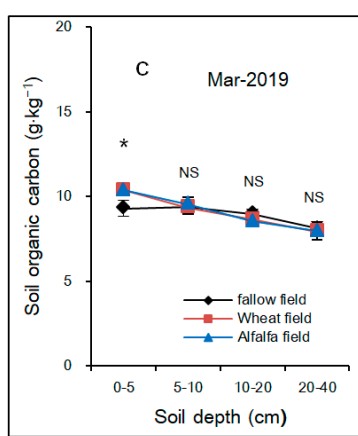

**Figure 2.** Changes in soil organic carbon under the three crop rotations. (**a**) shows the changes in topsoil organic carbon under the three crop rotations. (**b**) shows the changes in soil organic carbon in different soil layers under the three crop rotations in March 2016. (**c**) shows the changes in soil organic carbon in different soil layers under the three crop rotations in March 2019. Asterisks and NS indicate significant differences ($p < 0.05$) and insignificant differences between treatments, respectively.

Figure 3a shows that the TN in topsoil increased by 6.6%, 23.6%, and 53.4% at the end in the fallow, wheat, and alfalfa fields compared to the beginning. The fluctuation of TN in different soil layers was small at the beginning (Figure 3b), while at the end (Figure 3c), the TN showed a tendency to be larger in the upper layer and lower in the bottom layer, and the TN at a depth of 0–5 cm in the fallow, wheat, and alfalfa lands enhanced 24.1%, 17.0%, and 24.1% compared to that of 20–40 cm.

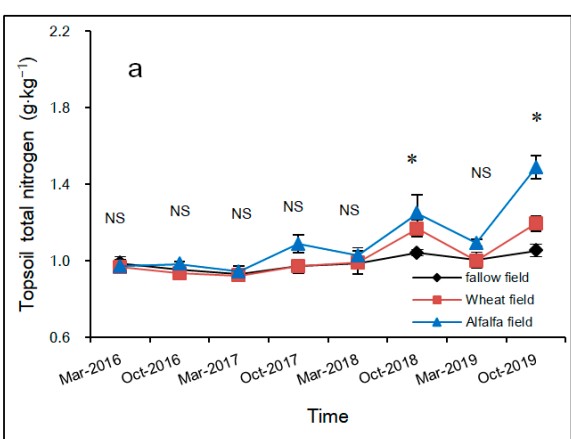
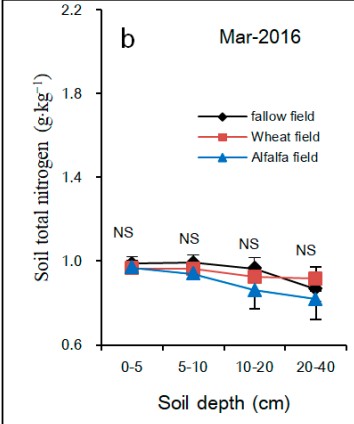
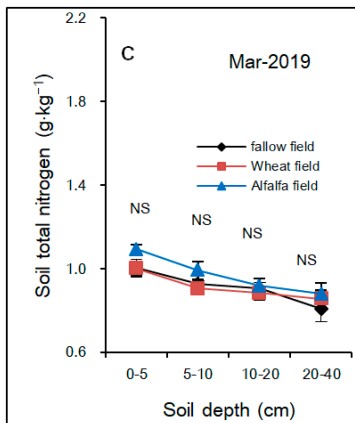

**Figure 3.** Changes in total nitrogen under the three crop rotations. (**a**) shows the changes in topsoil total nitrogen under the three crop rotations. (**b**) shows the changes in soil total nitrogen in different soil layers under the three crop rotations in March 2016. (**c**) shows the changes in soil total nitrogen in different soil layers under the three crop rotations in March 2019. Asterisks and NS indicate significant differences ($p < 0.05$) and insignificant differences between treatments, respectively.

During the whole experiment period, the changes of total phosphorus (TP) and total potassium (TK) in topsoil in the fallow, wheat, and alfalfa fields were little and only varied

by −1.9%, 3.5%, and 3.4%, and 3.3%, 2.7%, and −0.3% at the end compared with those at the beginning, respectively (Figures 4a and 5a). The variation of TP and TK in the soil profile was small during the experiment period (Figures 4b,c and 5b,c).

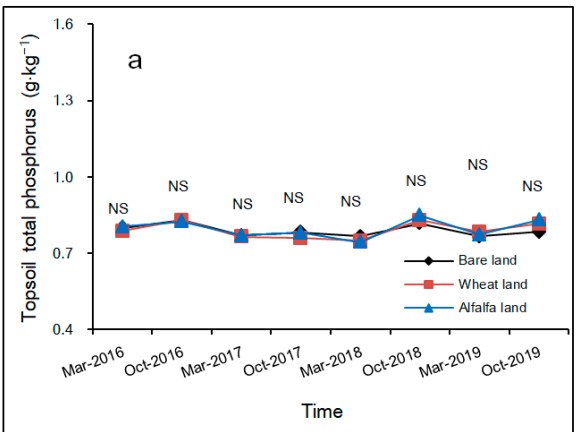
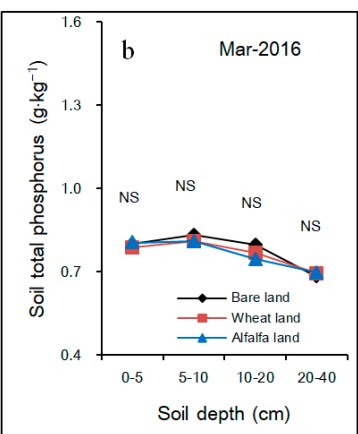
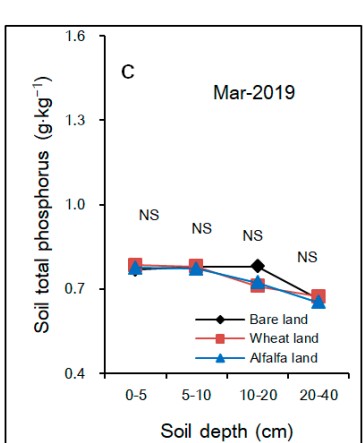

**Figure 4.** Changes in total phosphorus under the three crop rotations. (**a**) shows the changes in topsoil total phosphorus under the three crop rotations. (**b**) shows the changes in soil total phosphorus in different soil layers under the three crop rotations in March 2016. (**c**) shows the changes in soil total phosphorus in different soil layers under the three crop rotations in March 2019. NS indicates insignificant differences between treatments, respectively.

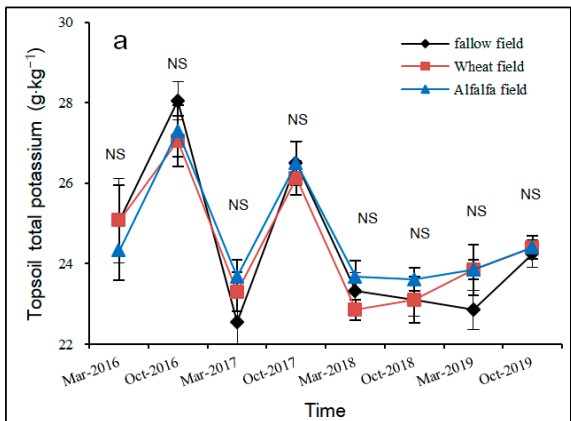
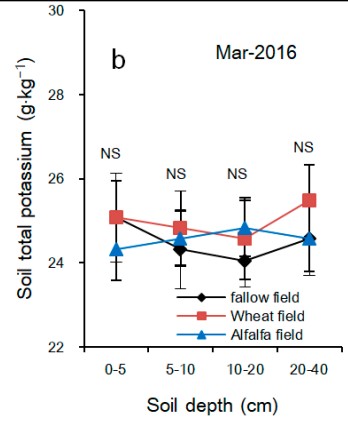
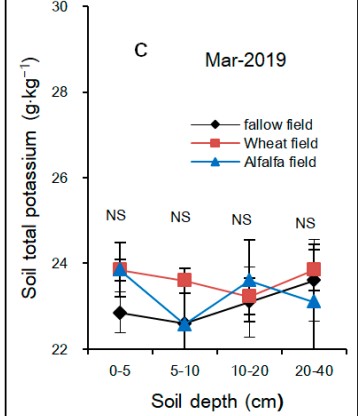

**Figure 5.** Changes in total potassium under the three crop rotations. (**a**) shows the changes in topsoil total potassium under the three crop rotations. (**b**) shows the changes in soil total potassium in different soil layers under the three crop rotations in March 2016. (**c**) shows the changes in soil total potassium in different soil layers under the three crop rotations in March 2019. NS indicates insignificant differences between treatments, respectively.

### 3.3. Changes in AN, AP, and AK under Three Crop Rotations

Crop rotations affected surface vegetation and further influenced the available nutrients in the soil. Figure 6a shows that the AN in topsoil decreased firstly and then increased with the extension of growing years, added by 12.6%, 38.7%, and 41.5% in the fallow, wheat, and alfalfa fields, respectively, at the end of the experiment compared with that in October 2017 (their lowest values), and the differences of AN in the rotations changed from no significance in the early to significance in the later. At the beginning of the experiment (Figure 6b), the AN varied greatly in different soil layers, but there was no significant difference in the same soil layer among the rotations. At the end of the experiment (Figure 6c), the AN showed a tendency to be larger in the upper layer and smaller in the lower layer; the AN of topsoil in the fallow, wheat, and alfalfa fields increased by 56.4%, 65.7%, and

100%, respectively, compared with that of the bottom layer; and the differences of AN in the 0–10 cm soil depth were significant among the rotations.

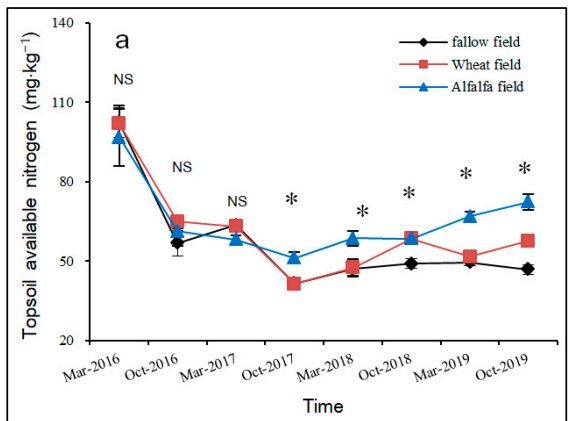 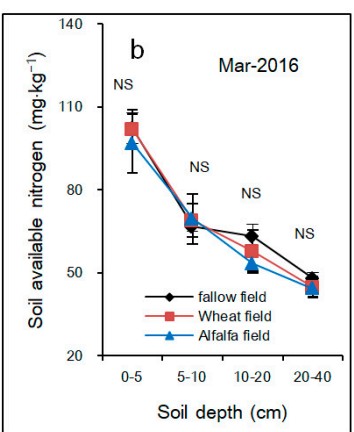 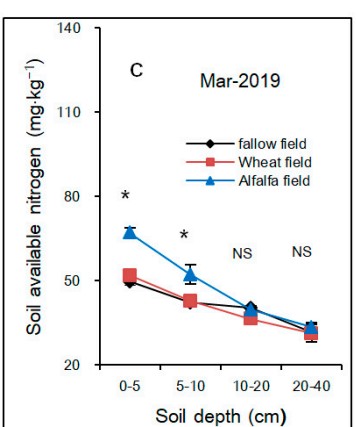

**Figure 6.** Changes in soil available nitrogen under the three crop rotations. (**a**) shows the changes in topsoil available nitrogen under the three crop rotations. (**b**) shows the changes in soil available nitrogen in different soil layers under the three crop rotations in March 2016. (**c**) shows the changes in soil available nitrogen in different soil layers under the three crop rotations in March 2019. Asterisks and NS indicate significant differences ($p < 0.05$) and insignificant differences between treatments, respectively.

Figure 7a shows that the AP in topsoil decreased firstly and then increased with the increase of growing years, increasing by −13.3%, 51.5%, and 41.4% in the fallow, wheat, and alfalfa fields at the end compared with that in October 2017 (the lowest values). At the beginning of the experiment (Figure 7b), the AP showed a tendency to be larger in the upper layer and smaller in the lower layer, but there was no significant difference in the same soil layer among the rotations. At the end of the experiment (Figure 7c), the AP had the same tendency as in the beginning; however, there were significant differences in the AP in the 5–20 cm soil layers among the rotations, and the AP in the fallow land was higher than that in wheat and alfalfa fields.

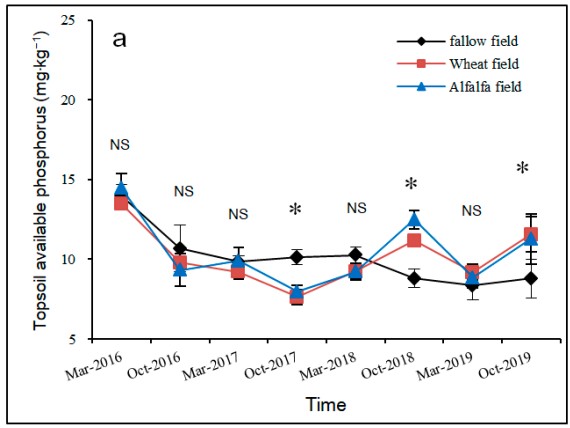 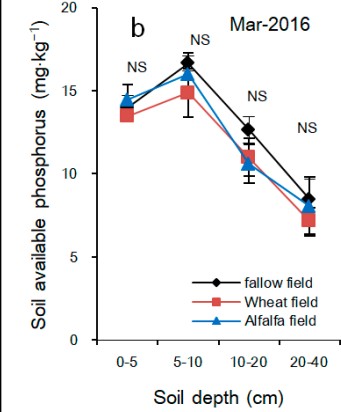 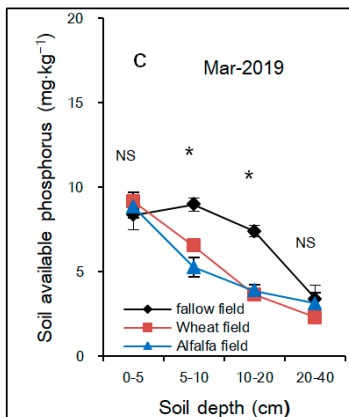

**Figure 7.** Changes in soil available phosphorus under the three crop rotations. (**a**) shows the changes in topsoil available phosphorus under the three crop rotations. (**b**) shows the changes in soil available phosphorus in different soil layers under the three crop rotations in March 2016. (**c**) shows the changes in soil available phosphorus in different soil layers under the three crop rotations in March 2019. Asterisks and NS indicate significant differences ($p < 0.05$) and insignificant differences between treatments, respectively.

The topsoil AK persistently increased throughout the test period, increasing by 90%, 167%, and 158% in the fallow, wheat, and alfalfa lands at the end compared with the

beginning, respectively (Figure 8a). At the beginning (Figure 8b), the AK in different soil layers changed little, while at the end (Figure 8c), it showed a tendency to be larger in the topsoil layer and smaller in the bottom layer, and the difference in the 5–20 cm soil layers was significant like AP.

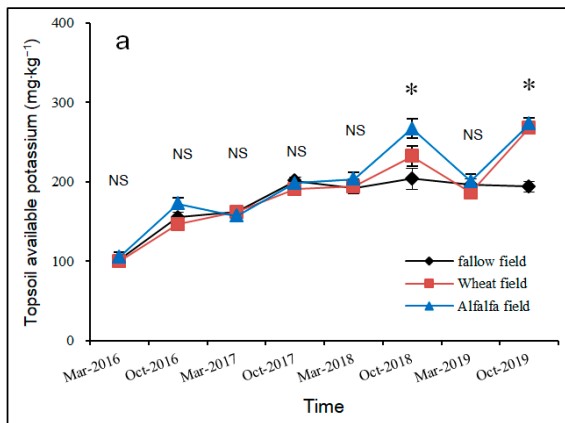
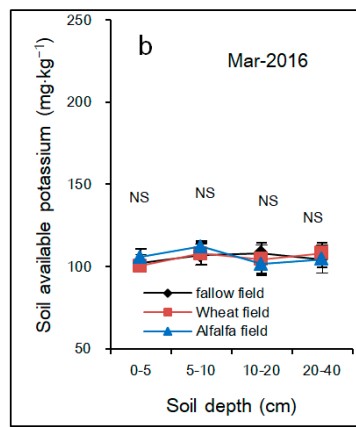
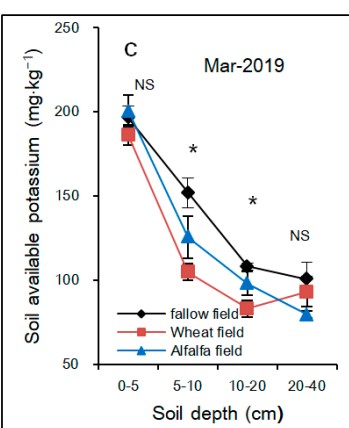

**Figure 8.** Changes in available potassium under the three crop rotations. (**a**) shows the changes in topsoil available potassium under the three crop rotations. (**b**) shows the changes in soil available potassium in different soil layers under the three crop rotations in March 2016. (**c**) shows the changes in soil available potassium in different soil layers under the three crop rotations in March 2019. Asterisks and NS indicate significant differences ($p < 0.05$) and insignificant differences between treatments, respectively.

### 3.4. Changes in Loss of Soil and Nutrients under Three Crop Rotations

Generally, the loss of soil and nutrients occurs from crop harvest to crop seedlings of the next year in semi-arid areas. This study therefore analyzed the effects of three crop rotations on soil and nutrient loss during three periods, i.e., from March 2016 to June 2017, March 2017 to June 2018, and March 2018 to June 2019 [3,34]. Crop rotations affected surface vegetation and further influenced the loss of soil and nutrients. As shown in Figure 9a, the soil loss in the fallow land was the most, varying from 1.60 to 1.98 kg·m$^{-2}$, with a mean of 1.76 kg·m$^{-2}$, while that in wheat and alfalfa fields varied from 1.35 to 1.62 kg·m$^{-2}$ and 0.86 to 1.12 kg·m$^{-2}$, with averages of 1.44 and 0.98 kg·m$^{-2}$. The differences among the rotations were significant. Figure 9b shows that the loss of SOC in the fallow, wheat, and alfalfa lands varied from 137.4 to 168.8 kg·hm$^{-2}$, 120.3 to 146.1 kg·hm$^{-2}$, and 79.4 to 115.7 kg·hm$^{-2}$, respectively, with means of 151.8, 135.5, and 95.1 kg·hm$^{-2}$, and the loss in the wheat and alfalfa fields decreased by 10.7% and 37.4% than that in the fallow field. The loss of TN in the wheat and alfalfa fields was reduced by 18.1% and 40.9% compared to the fallow field (Figure 9c). The losses of TP and TK in the wheat and alfalfa fields were 11.0, 7.5 kg·hm$^{-2}$, and 353.9, 240.9 kg·hm$^{-2}$, respectively, and decreased by 18.5%, 44.4%, and 16.8%, 43.3% compared to the fallow field, respectively (Figure 9d,e).

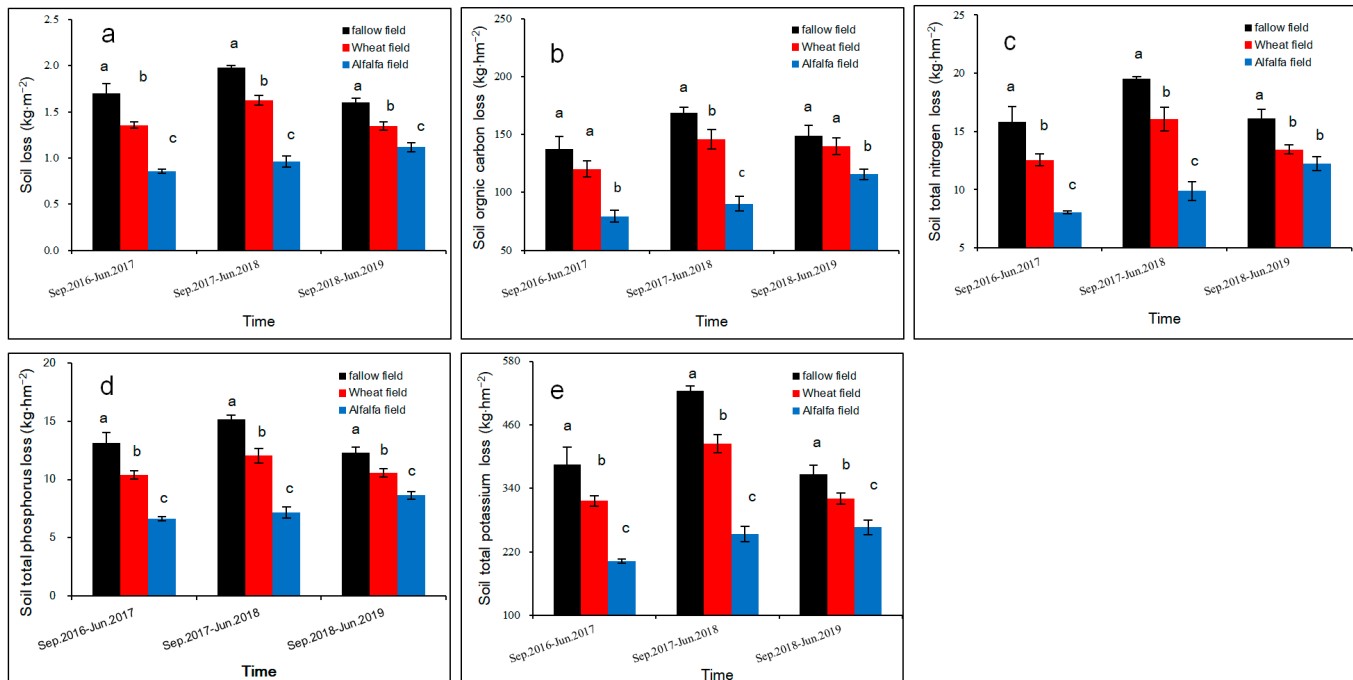

**Figure 9.** Changes in loss of soil and nutrients under the three crop rotations. (**a**) shows the changes in soil loss under the three crop rotations. (**b**) shows the changes in soil organic carbon loss under the three crop rotations. (**c**) shows the changes in soil total nitrogen loss under the three crop rotations. (**d**) shows the changes in soil total phosphorus loss under the three crop rotations. (**e**) shows the changes in soil total potassium loss under the three crop rotations. Different lowercase letters in the figure indicate significant differences ($p < 0.05$) between treatments.

*3.5. Correlation and Regression Analysis among Different Determined Indexes*

A statistical analysis of different determined indexes was performed using the data in March 2019. Table 1 indicates that there were significant positive correlations between the surface vegetation characteristics, among which the correlation between the coverage and surface biomass was the strongest (0.973), followed by the coverage and height (0.878). There were positive correlations among the soil nutrients; however, only the correlation between SOC and TN was significant. There were positive correlations between the soil nutrients and vegetation characteristics, among which the correlations of the SOC, TP, and TK with the height were the strongest, reaching 0.588, 0.173, and 0.531, respectively, while that of the TN with the coverage was the largest (0.600). There were significant positive correlations between soil loss and nutrient loss. The losses of soil, TN, and TP with the height had the strongest negative correlations with the coefficients of −0.917, −0.821, and −0.899, while the correlations between the SOC loss and the surface biomass, the TK loss, and the coverage were the strongest, with r values of −0.755 and −0.830, respectively. In addition, there was no significant negative correlation between soil nutrients and the loss of soil nutrients.

**Table 1.** The correlations between surface vegetation characteristics, soil nutrients, and loss of soil and nutrients.

| Factor | Height | Surface Biomass | SOC | TN | TP | TK | Soil Loss | SOC Loss | TN Loss | TP Loss | TK Loss |
|---|---|---|---|---|---|---|---|---|---|---|---|
| Coverage | 0.878 ** | 0.973 ** | 0.441 | 0.600 * | 0.076 | 0.234 | −0.846 ** | −0.732 ** | −0.689 * | −0.846 ** | −0.830 ** |
| Height | | 0.799 ** | 0.588 * | 0.475 | 0.173 | 0.531 | −0.917 ** | −0.717 ** | −0.821 ** | −0.899 ** | −0.826 ** |
| Surface biomass | | | 0.346 | 0.574 | 0.039 | 0.185 | −0.817 ** | −0.755 ** | −0.668 * | −0.822 ** | −0.814 ** |

**Table 1.** *Cont.*

| Factor | Height | Surface Biomass | SOC | TN | TP | TK | Soil Loss | SOC Loss | TN Loss | TP Loss | TK Loss |
|---|---|---|---|---|---|---|---|---|---|---|---|
| SOC | | | | 0.587 * | 0.049 | 0.438 | −0.517 | −0.006 | −0.279 | −0.528 | −0.417 |
| TN | | | | | 0.141 | 0.161 | −0.519 | −0.267 | −0.082 | −0.503 | −0.498 |
| TP | | | | | | 0.043 | −0.168 | −0.172 | −0.122 | 0.029 | −0.150 |
| TK | | | | | | | −0.35 | −0.135 | −0.299 | −0.344 | −0.099 |
| Soil loss | | | | | | | | 0.859 ** | 0.893 ** | 0.980 ** | 0.966 ** |
| SOC loss | | | | | | | | | 0.871 ** | 0.828 ** | 0.882 ** |
| TN loss | | | | | | | | | | 0.879 ** | 0.872 ** |
| TP loss | | | | | | | | | | | 0.950 ** |

Note: The soil nutrients (SOC, TN, TP, and TK) are the nutrients in the 0–5 cm soil layer, and the loss of soil and nutrients (SOC, TN, TP, and TK) are the loss of soil and nutrients in the 0–5 cm soil layer. * indicates that the correlation is significant at the level of 0.05, and ** indicates that the correlation is significant at the level of 0.01.

Due to the correlations between soil nutrients and loss of soil and nutrients, with the coverage or height of vegetation characteristics being the most important, linear regression is performed with coverage and height as independent variables. As shown in Table 2, the SOC in topsoil would increase by 0.076 $g \cdot kg^{-1}$ for every 1 cm increase in vegetation height, the TN would increase by 0.003 $g \cdot kg^{-1}$ for every 1% increase in vegetation coverage, and their $R^2$ values were 34.6% and 36.0%, respectively. The loss of topsoil, TN, and TP would increase by 0.035 $g \cdot m^{-2}$, 0.283, and 0.262 $kg \cdot hm^{-2}$ for a 1 cm decrease in the vegetation height; the loss of SOC and TK would increase by 0.933 and 2.697 $kg \cdot hm^{-2}$ for a 1% decrease in the vegetation coverage; and the $R^2$ of these fitting equations was more than 53.6%, among which the $R^2$ between the soil loss and height was as high as 84.1%.

**Table 2.** The quantitative relations between the soil nutrients, the loss of soil and nutrients, and the vegetation coverage or height.

| Factor | Coverage (%) | | | Height (cm) | | |
|---|---|---|---|---|---|---|
| | Fitted Equation | $R^2$ | *p* | Fitted Equation | $R^2$ | *p* |
| SOC $(g \cdot kg^{-1})$ | y = 9.707 + 0.002x | 0.194 | 0.151 | y = 9.461 + 0.076x | 0.346 | 0.044 |
| TN $(g \cdot kg^{-1})$ | y = 0.992 + 0.003x | 0.36 | 0.039 | y = 0.989 + 0.006x | 0.225 | 0.119 |
| TP $(g \cdot kg^{-1})$ | y = 0.774 + 0.000x | 0.006 | 0.814 | y = 0.771 + 0.001x | 0.03 | 0.591 |
| TK $(g \cdot kg^{-1})$ | y = 23.31 + 0.015x | 0.055 | 0.465 | y = 22.87 + 0.090x | 0.282 | 0.076 |
| Soil loss $(g \cdot m^{-2})$ | y = 1.526 − 0.012x | 0.716 | 0.001 | y = 1.607 − 0.035x | 0.841 | 0.000 |
| SOC loss $(kg \cdot hm^{-2})$ | y = 147.8 − 0.933x | 0.536 | 0.007 | y = 152.1 − 2.384x | 0.514 | 0.009 |
| TN loss $(kg \cdot hm^{-2})$ | y = 15.18 − 0.091x | 0.475 | 0.013 | y = 15.95 − 0.283x | 0.674 | 0.001 |
| TP loss $(kg \cdot hm^{-2})$ | y = 11.81 − 0.094x | 0.716 | 0.001 | y = 12.39 − 0.262x | 0.808 | 0.000 |
| TK loss $(kg \cdot hm^{-2})$ | y = 355.3 − 2.697x | 0.689 | 0.001 | y = 368.4 − 7.004x | 0.682 | 0.001 |

Note: The soil nutrients (SOC, TN, TP, and TK) are the nutrients in the 0–5 cm soil layer, and the loss of soil and nutrients (SOC, TN, TP, and TK) are the loss of soil and nutrients in the 0–5 cm soil layer.

## 4. Discussion

### 4.1. Effects of Various Crop Rotations on Surface Vegetation Characteristics

Crop rotation significantly influenced the aboveground vegetation characteristics. Although wheat and alfalfa were simultaneously seeded in spring at the beginning of the trial, the growing rate of alfalfa seedlings was slower compared to that of wheat seedlings. This could be because the biomass of alfalfa seed is smaller than that of wheat seed; the alfalfa seedling obtains fewer nutrients from its seed compared to wheat seeding, so the alfalfa seedling grew slower than the wheat seedling. Therefore, the surface vegetation coverage, height, and biomass in the alfalfa field were smaller than those in the wheat field in the first growth season. After wheat was harvested, the stubble retained on the surface was blown and rained, resulting in the surface vegetation characteristics continuously decreasing. Although mowed 2–3 times during the growing season and no irrigation after wheat harvest, alfalfa could still grow by absorbing rainfall; this made the vegetation

characteristics in the alfalfa field greater than those in the wheat field in October. In the spring of other years, the growing rate of alfalfa seedlings grew significantly faster compared to that of wheat seedlings. The reasons are that alfalfa, as a perennial forage, germinated relatively early in spring, while the growing rate of wheat seedlings was slowed down and influenced by seeding in spring and continuous cropping. The plants in the fallow field were negligible because of sprayed herbicides and manual weeding. Therefore, the aboveground vegetation characteristics in the three crop rotations in spring and autumn had a significant difference, and their rank order was fallow field < wheat field < alfalfa field [29].

*4.2. Effects of Various Crop Rotations on the Loss of Soil and Nutrients*

Studies indicated that land use variation and agricultural intensification are the major causes of habitat degradation [11,14]. Soil erosion by wind was very sensitive to land use conversion caused by anthropogenic activities [35]. Land use patterns influenced surface vegetation cover and further affected soil loss [3]. The reasons are that aboveground vegetation or stubble can increase the surface roughness, avoid the exposure of soil to wind, absorb and disperse wind momentum, and decrease the momentum transfer between the ground materials and air flow [36–38]. The control of surface vegetation or stubble on wind erosion depends on its amount and duration [38]. To effectively control wind erosion, the surface vegetation coverage should exceed the critical vegetation coverage (60%) [25]. The land use pattern influenced the surface vegetation cover and further affected soil physical and chemical properties [17,39]. The surface vegetation cover could increase moisture and the starting wind speed in the topsoil and accordingly reduce soil loss [3]. Increasing vegetation cover could reduce topsoil salinity [29,40], while high salinity in topsoil could cause soil desertification [41,42].

This study found that during the wind erosion period, the soil loss in the fallow field was the greatest, followed by the wheat and alfalfa fields. Statistical analysis further showed that the correlation coefficient between the soil loss and surface vegetation characteristics was negative, which indicates that the soil loss would increase when the surface vegetation decreased [38]. The reasons are that, firstly, the topsoil particles were in a discrete state under the action of high salt ion concentrations, such as $Na^+$ [23,29,42]. Secondly, the freeze-thaw process changed the physical and chemical properties of soil, such as increased salinity and dispersion in topsoil, and aggravated the erodibility of soil [34]. Thirdly, the weather in the region was characterized by intense sunshine, scarce precipitation, and wind during the wind erosion period. In the experimental region, wind erosion often happens from March to May of each year. Due to little surface vegetation and high topsoil salinity [29], the soil loss in the fallow field was the greatest. Due to the amounts of litter and stubble retained and the early germination of perennial alfalfa in spring under low soil salinity [29], the alfalfa field indicated strong resistance to wind erosion. In addition, because the vegetation coverage in the alfalfa field was less than the critical coverage, a little soil loss occurred in the field, but the loss was the lowest among the three treatments and decreased by 44.3% compared to the fallow field. Although the amount of surface vegetation was low, the wheat field still exhibited some resistance to wind erosion, and the soil loss was lower compared to the fallow land (a reduction of 18.2%) [3]. Therefore, to reduce wind erosion in arid and semi-arid irrigated regions, it is essential to retain some crop stubble on the surface during the non-growing season, and conventional tillage should be avoided [38].

Correlation analysis indicated that surface vegetation characteristics are the most important factors affecting soil nutrient loss. As there were significant negative correlations between soil loss and vegetation characteristics and few plants on the surface, the soil loss in the fallow field was the largest, which resulted in the most loss of nutrients. Although the topsoil nutrients were high, the nutrient loss in the alfalfa field was the smallest among the three rotations because of the smallest soil loss. For example, the loss of SOC, TN, TP, and TK was reduced by 37.4%, 40.9%, 44.4%, and 43.3%, respectively, compared to the

fallow field. The topsoil loss in the wheat field was the second, and the loss of SOC, TN, TP, and TK decreased by 10.7%, 18.1%, 18.5%, and 16.8%, respectively, compared with that in the fallow field. It can be seen from the above results that the high nutrient contents in topsoil do not mean the large nutrient loss caused by wind, and the nutrient loss is mainly affected by the amount of surface vegetation and soil loss.

*4.3. Effects of Various Crop Rotations on Soil Carbon, Nitrogen, Phosphorus, and Potassium*

Crop rotation influenced the surface vegetation and further affected the soil nutrients [21,43]. Yang et al. [11] found that rice-tiny vetch (used as green manure in situ) and rice-fallow rotations increased SOC, macrofaunal biodiversity, and trophic structure after 3-year rotations. Crop-forage rotation maintained or increased SOM, and the SOM showed a significant stratified distribution in no-tillage systems, with greater concentrations in the uppermost layers of silt loam (0−10 cm) and clay loam soils (0−30 cm); rotating perennial forages and no-tillage in corn-based agricultural systems could increase organic resource abundance [12]. With the extension of cultivation ages, the SOM and TN of different soil layers in the alfalfa field gradually increased [23]. However, Deiss et al. [12] also found that under no-tillage, corn soybean rotation diminished SOM accrual compared to continuous corn or corn forage rotations. The fine soil particles contained abundant nutrients, and the loss rate of nutrients caused by wind was greater than that of fine particles [33]. Therefore, soil nutrient variation in arid and semi-arid areas is not only affected by the input of litter and roots but also closely related to the loss of fine soil particles.

This study showed that with the extension of growing years, the increasing rate of topsoil nutrients all showed increasing trends and exhibited obvious differences among the three rotations in the later period. For example, the SOC in the alfalfa, wheat, and fallow fields in the autumn of the fourth year increased by 64.5%, 33.1%, and 21.9%, respectively, compared to the spring of the first year. Possible reasons are as follows: First, the decomposition of the litter, stubble, and dead roots of crops increased soil nutrients. Second, soil nutrient loss during the wind erosion period had obvious differences among the rotations. Third, the amount of organic matter entering the soil was different under the three rotations. These reasons resulted in the highest increasing rate of SOC and TN in the alfalfa field, followed by wheat and fallow fields. The TP and TK changed little or even showed a decreasing trend due to no fertilization or crop consumption during the experiment period. The AN and AP first decreased and then increased. This might be the excessive application of fertilizers such as ammonium dihydrogen phosphate and urea before the experiment; residual soluble nitrogen and phosphorus nutrients were retained in the soil, resulting in the high concentration of AN and AP at the beginning; along with crop consumption, the AN and AP gradually decreased and reached the minimum. With the extension of crop growth, the effect of crop roots on insoluble nutrients in soil increased, and the AN and AP showed increasing trends again [23]. In addition, the symbiotic $N_2$ fixation of leguminous crop roots and rhizobia also promoted an increase in soil nitrogen [11]; soil organic carbon in the autumn was higher than that in the spring of the following year, which also reflected the negative effect of wind erosion on soil nutrients (Figure 2). According to the variation tendency of SOC and TN, it indicates that perennial plants, such as alfalfa, could significantly increase soil nutrients after two years of growth [21,23], while the increase rate of soil nutrients in the fallow field was relatively slow and even decreased under severe wind erosion. Of course, the increase in topsoil nutrients under alfalfa rotation is not unlimited. With the increase in growing years, the topsoil nutrient contents under alfalfa rotation will be in balance under the combined action of wind erosion, plant absorption, and organic matter input and reach their maximum.

**5. Conclusions**

This study investigated the changes in surface vegetation characteristics, soil nutrients, and loss of soil and nutrients under different crop rotations in a semi-arid area and analyzed their changes and numerical relationships. The coverage, height, and biomass

of the aboveground vegetation in the three crop rotations in spring and autumn had a significant difference, and the rank order was fallow field < wheat field < alfalfa field. Alfalfa rotation could significantly increase SOC, TN, AP, and AK concentrations after two years of cultivation but had little impact on TP and TK. The loss of soil and nutrients (SOC, TN, TP, and TK) under alfalfa rotation was the least, followed by wheat and fallow rotations. There were significant negative correlations between the surface vegetation characteristics and the loss of soil and nutrients. As far as improving soil fertility is concerned, alfalfa rotation with stubble retained on the surface can reduce the loss of soil and nutrients and significantly increase soil nutrients after two years of cultivation, which is conducive to the sustainability of agroecosystems.

**Author Contributions:** All authors contributed to the study conception and design. Conceptualization: A.L., X.T. and T.G.; Methodology: A.L., S.C. and T.G.; Writing—original draft preparation: A.L., X.T. and T.G.; Writing—review and editing: Y.W., S.C. and T.G.; Funding acquisition: A.L., S.C. and T.G.; Investigation: A.L., X.T., S.C. and T.G.; Validation: A.L. and T.G. All authors have read and agreed to the published version of the manuscript.

**Funding:** This research was funded by the National Natural Science Foundation of China, grant number 31560185 and 31860176; the Science and Technology Plan Project of Gansu Province, grant number 21JR7RA539 and 20YF3FA037; the Key Research and Development Program of Shaanxi, grant number 2020ZDLSF06-06.

**Institutional Review Board Statement:** Not applicable.

**Informed Consent Statement:** Not applicable.

**Data Availability Statement:** Data are available from the authors upon reasonable request.

**Acknowledgments:** The authors would like to thank the Xi'an Key Laboratory of plant stress physiology and ecological restoration technology, and the Key Laboratory for Ecological Remediation and High-quality development of the Qinling Mountains in the Upper and Middle Reaches of the Yellow River laboratories for their assistance once again.

**Conflicts of Interest:** The authors declare no conflict of interest.

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
