# Peer review of "Effects of Alfalfa Crop Rotation on Soil Nutrients and Loss of Soil and Nutrients in Semi-Arid Regions"

_sustainability, doi:10.3390/su152015164_

Round 1

Reviewer 1 Report

1.  What I could understand that there were only two crop rotations i.e. rape-alfalfa, rape-wheat and rape-fallow may not considered as rotation. Even if authors claimed as 3 rotations, the experiment does not satisfy the necessary design requirement of 12 degree of freedom. It is though not sacrosanct, but a guiding principle in agronomic experiment for minimizing the errors. With four replications, and three treatments, df is just 6. The experiment is acceptable but premise of recommendation may be considered weak.

2. Have the authors maintained the same rotations in the same plots in all four years. This should be obvious but a mention may be there.

3. In table 1, the critical value @1% and @5% should be mentioned at the footnote.

4. Losses of soils and nutrients are obvious trends

5. The increasing trend of available N P and K over the years is encouraging. The author may indicate the trends of the increase. It cannot go on increasing in which year the increment will reach the maximum, if possible such projection will enhance the value to the analyses.

Reviewer 2 Report

My comments are intended to improve your research work.

Reviewer 3 Report

In semi-arid region, the conservation agriculture such as crop rotation can reduce soil erosion and nutrient loss. The authors compared the effects of various crop rotations on soil nutrients by a 4-year field experiment. The results showed that crop-alfalfa rotation can reduce the loss of soil and nutrients and increase soil nutrients in semi-arid area. Long term experiments can provide very useful reference basis for agricultural production, sustainable development of agriculture, and environmental protection. The article is well written, easy to read, articulate, and can be accepted for publication after addressing the following issues.

1. Introduction

Line 42, “Crop rotation induces several crop species” What does means?

Line 57-59, need references/

It was suggested to show more related studies. Not only in China. The studies in the worldwide was also suggested to show in the introduction.

2. Materials and Methods

Line 107, “In spring 2016” Please indicate the specific month.

“2.2 Experimental Design and Sampling Processing” This section includes experimental design and sample analysis, and it is suggested to be divided into two sections

There are multiple analytical methods for the same physical and chemical properties, and it is recommended to add a description of the soil sample analysis method.

The sentences in Line 137-140, “The loss of ------during this period” Suggest moving to a suitable location or expressing this content in the results section

3. Results

All Figures, Add a more detailed description in the figure legend (Such as The Meaning of Asterisks or NS, lowercase in statistical analysis ), so that readers can understand the meaning of the diagram without looking at the main text.

Suggest unifying fonts in figures

In the results, the authors used Various Crop Rotations (Such as in line 202), Only three Crop Rotations were conducted in this experiment, It was suggested to use three Crop Rotations. Normally, various means many

Minor editing of English language required

Round 2

Reviewer 1 Report

Can be accepted at the present form now.

Reviewer 2 Report

The document improved with the changes made